# Doubly N-confused and ring-contracted [24] hexaphyrin Pd-complexes as stable antiaromatic N-confused expanded porphyrins

Fuying Luo[1,2], Le Liu[1,2], Han Wu[1], Ling Xu[1], Yutao Rao[1], Mingbo Zhou[1], Atsuhiro Osuka[1] & Jianxin Song[1] ✉

As isomers of the regular porphyrins, N-confused porphyrins have attracted extensive attention of chemists because of their unique chemical structures, chemical reactivities, and physical properties, which result in their promising applications in the fields of catalytic chemistry, biochemistry and material science. Typically, N-confused porphyrins are synthesized via acid catalyzed condensation and following oxidation during which lactams are often formed as the byproducts. Here we report doubly N-confused and ring-contracted [24] hexaphyrin(1.1.0.1.1.0) mono- and bis-Pd-complexes as stable antiaromatic N-confused expanded porphyrins, which are synthesized through Pd-catalyzed Suzuki-Miyaura coupling of 1,14-dibromotripyrrin. These macro-cycles show a paratropic ring currents, an ill-defined Soret band, a red-shifted weak absorption tail, and a small HOMO-LUMO gap. NBS bromination of the bis Pd-complex give its mono- and dibromides regioselectively, which are effectively used to synthesize a [24]hexaphyrin dimer and a Ni[II] porphyrin-[24] hexaphyrin-Ni[II] porphyrin triad, respectively.

In the chemistry of porphyrinoids, expanded porphyrins[1–5] and N-confused porphyrins (NCPs)[6–9] are two important newcomers, which have contributed to highlighting the glorious potentials of porphyrinoids. In 2003, Furuta et al. have combined these two porphyrinoids by synthesizing expanded NCPs such as doubly N-confused [26]dioxohexaphyrins(1.1.1.1.1.1) **1** and doubly N-confused and ring-contracted [26]dioxohexaphyrins(1.1.1.1.1.0) **2** (Fig. 1)[10–14]. These molecules were synthesized by acid-catalyzed condensation reactions and the lactam units were formed via facile oxidation of the N-confused pyrroles. The oxidized N-confused pyrrole units in **1** and **2** have been shown to be quite effective for the coordination of transition metal ions, allowing for the synthesis of various metal complexes. On the other hand, attempts to synthesize doubly N-confused non-oxo [26]hexaphyrin metal complexes ended in very low yields[15]. In addition, these synthetic methods provided only thermodynamically stable aromatic products. As a rare example, Furuta et al. reported that electrochemical

oxidation of **2Cu** led to the generation of 24π-antiaromatic species[14], but antiaromatic expanded NCP has been never isolated before. In this work, we report the serendipitous synthesis of *d*oubly *N*-confused and *r*ing-*c*ontracted (DNCRC) [24]hexaphyrin(1.1.0.1.1.0) Pd[II] complexes **8** and **9** by Pd-catalyzed self-coupling reaction of α,α-dibromo-tripyrrin **3b**.

## Results

With the aim to synthesize *m*-benziporphyrin(1.1.0.0) **5**, we attempted the cyclization reaction of α,α'-diboryltripyrrane **3a** with 1,3-dibromo-4-methoxybenzene under the usual Suzuki−Miyaura coupling conditions (Pd₂(dba)₃, X-phos, Cs₂CO₃, CsF, and toluene/DMF). In addition to the target **5**[16] which was actually obtained in 5% yield, we isolated dark green product **6** in ca. 1% (Fig. 2). Fortunately, the structure of **6** has been revealed by X-ray analysis to be a symmetric and planar [18] triphyrin(5.1.1) with a small mean-plane deviation (MPD) of 0.26 Å

[1]Key Laboratory of Chemical Biology and Traditional Chinese Medicine Research (Ministry of Education of China), Key Laboratory of the Assembly and Application of Organic Functional Molecules of Hunan Province, College of Chemistry and Chemical Engineering, Hunan Normal University, 410081 Changsha, China. [2]These authors contributed equally: Fuying Luo, Le Liu. ✉e-mail: jxsong@hunnu.edu.cn

**Fig. 1 | Structures of N-confused hexaphyrins.** Structures of doubly N-confused [26]dioxohexaphyrins(1.1.1.1.1.1) **1**, doubly N-confused and singly ring-contracted [26]dioxohexaphyrins(1.1.1.1.1.0) **2**, and its Cu[II] complex **2Cu**.

**Fig. 2 | Synthesis of compounds 5 and 6.** Dba = dibenzylideneacetone, Mes = 2,4,6-trimethylphenyl, Bpin = pinacolatoboryl, DDQ = 2,3-dicyano-5,6-dichlorobenzoquinone.

(Fig. 3a, b). The carbonyl oxygen atom is hydrogen bonded with the two adjacent pyrrolic NH groups and the C=O bond length is 1.288(7) Å, being distinctly longer than those of usual ketones (1.23 Å)[17, 18]. This structural feature suggests a significant contribution of a dipolar resonance state. The $^1H$ NMR spectrum of **6** shows a singlet at 9.74 ppm due to the vinylic proton, two doublets at 8.36 and 8.08 ppm ($J = 4.0$ Hz) and a singlet at 8.31 ppm due to the pyrrolic protons, and signals due to the NH protons at 3.89 (2H) and −4.13 (1H) ppm, suggesting a distinct diatropic ring current, reflecting its $18\pi$-electronic circuit. The absorption spectrum of **6** shows a split Soret-like band at 419 and 444 nm and vibronic-structured Q-like bands as characteristic features of aromatic porphyrinoids (Fig. 4). Further, **6** displays vibronic-structured fluorescence at 652 and 721 nm with a high quantum yield of $\Phi_F = 0.48$. These data strongly indicate that **6** is an aromatic tripyrrolic porphyrinoid. Actually, the calculated NICS values are negative (−12.15 ppm) inside the macrocycle.

Since **6** can be regarded as a cyclized product of **3a** and dibenzalacetone, we examined the reaction of only **3a** under the same conditions, which actually gave **6** in 5–8% yield. In the next step, we examined the reaction of **3a** with dibenzalacetone and found that **6**

was obtained in 8.8% under conditions (3 eq. dibenzalacetone, 0.4 equiv PPh$_3$, 2 mol% Pd$_2$(dba)$_3$, 2 eq. Cs$_2$CO$_3$, CsF). In continuation of this study, we examined the reaction of 1,14-dibromotripyrrin **3b** under the similar Suzuki−Miyaura coupling conditions (X-phos, Cs$_2$CO$_3$, CsF, Pd$_2$(dba)$_3$, toluene, DMF, reflux, 48 h), and found quite unexpectedly that [24]hexaphyrin Pd-complexes **8** (9%) and **9** (10%) were obtained along with **6** (1%) and tripyrrolic compound **7** (2%) (Fig. 5). The structure of **7** has been revealed by X-ray analysis (Fig. 3c, d) to be a coupling product of **3** and dibenzalacetone.

The parent ion peaks of **8** and **9** were observed at m/z = 1120.2568 (calcd for (C$_{64}$H$_{54}$N$_6$Pd$_2$)$^+$ = 1120.2504 ([$M$]$^+$)) and at m/z = 1030.3687 (calcd for (C$_{64}$H$_{56}$N$_2$Pd)$^+$ = 1030.3566 ([$M$]$^+$)) by high-resolution MALDI-TOF mass measurement. Both the structures of **8** and **9** have been revealed to be *d*oubly *N*-*c*onfused and *r*ing-*c*ontracted (DNCRC) [24]hexaphyrins(1.1.0.1.1.0) by X-ray analysis (Fig. 6). Product **8** is a nonoxo DNCRC [24]hexaphyrin, displaying a very planar structure with a small MPD (0.04 Å), in which the two Pd metals are coordinated with the three pyrrolic nitrogen atoms and one $\alpha$-carbon of the N-confused pyrrole with a bond distance of 2.002(4) Å for Pd1−N1, 2.027(5) Å for Pd1−N2, 1.959(5) Å for Pd1−N3, and 1.972(5) Å for Pd1−C.

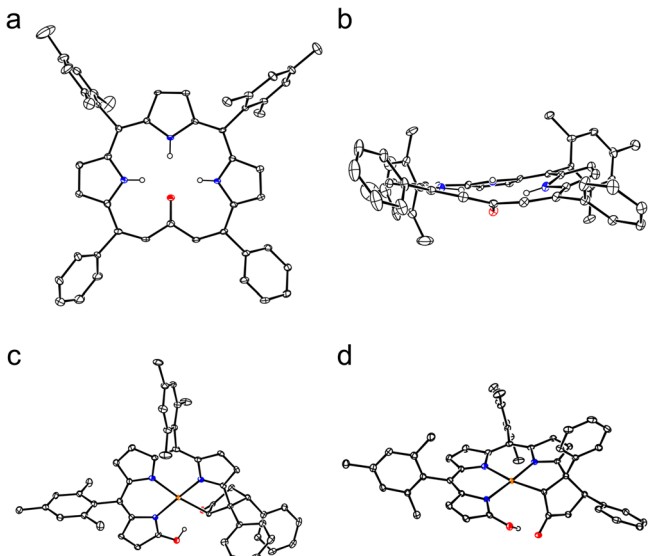

**Fig. 3 | X-ray crystal structures of 6 and 7. a** Top view of **6**, **b** side view of **6**, **c** top view of **7**, **d** side view of **7**. Ellipsoids are drawn at the 30% probability level. All hydrogen atoms except those connected to N and O atoms are omitted for clarity. Carbon atom, black ellipsoid; nitrogen atom, blue; palladium atom, orange; oxygen atom, red; hydrogen atom small black ball (These instructions are omitted in the following figures for clarity).

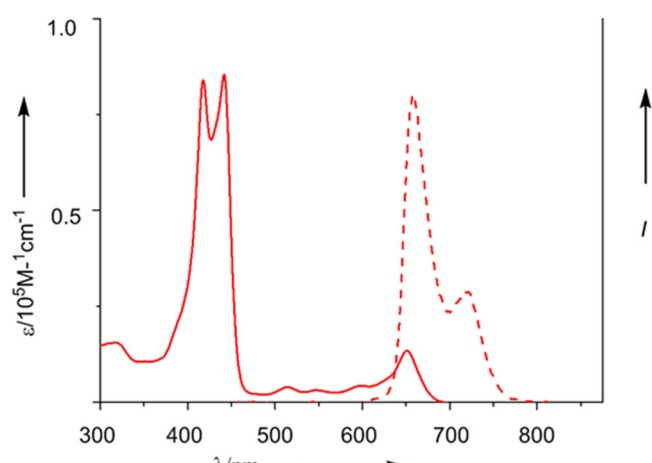

**Fig. 4 | UV–Vis absorption spectrum and fluorescence spectrum of 6 in $CH_2Cl_2$.** Solid line, absorption spectrum; dash line, fluorescence spectrum.

As compared with the previously reported $Pd^{II}$ complexes of NCPs that showed bond distances of Pd–C (2.002–2.003 Å) and Pd–N (2.003 Å)[19–21], Pd1–N2 and Pd1–C bond lengths are distinctly shorter. The $^1H$ NMR spectrum of **8** shows signals due to the β-pyrrolic protons as a singlet at 4.01 ppm and two doublets at 4.85 and 5.10 ppm ($J$ = 4.5 Hz) and at 5.25 and 5.30 ppm ($J$ = 4.5 Hz), indicating a weak but distinct paratropic ring current. The absorption spectrum of **8** is ill-defined, showing peaks at 396, 502, 618, 687, 763, 1015, and 1176 nm. The HOMA value was calculated to be 0.51 based on the crystal-lographic data, and the NICS value at the center of the molecule was calculated to be 5.41, indicating the weak antiaromatic character of **8**. In addition, the ACID plot (see Supplementary Fig. 58) of complex **8** displayed a continuous paratropic ring current, being consistent with the NICS(0) value.

The structure of **9** has been determined by X-ray diffraction analysis to be a DNCRC keto tautomer, in which one α-position of the N-confused pyrrole is oxidized. Unfortunately, the serious disorder impeded a detailed analysis of the structure. The $^1H$ NMR spectrum of **9** displays two signals due to the inner NH protons at 21.89 and 19.85 ppm and those in the range of 5.22–3.79 ppm due to the pyrrolic β-protons, clearly indicating its paratropic ring current and thus anti-aromatic character. Judging from these chemical shifts, it is con-ceivable that the antiaromatic character of **9** is slightly larger as compared with **8**. The NICS value of **9** was calculated to be 10.43, which is larger than that of **8**. Importantly, **8** and **9** are stable antiaromatic N-confused expanded porphyrins.

Hexaphyrin **9** has been demonstrated to be a nice platform to form bis-metalated complexes. Actually, complexes **9Pd**, **9Ni**, and **9Cu** were readily obtained in 80%, 95%, and 85% upon complexation with $Pd^{II}$, $Ni^{II}$, and $Cu^{II}$ ions, respectively. The $^1H$ NMR spectra of **9Pd** and **9Ni** show signals due to the peripheral β-protons in the range of 5.40–4.09 and 5.20–3.77 ppm, respectively, revealing paratropic rings current comparable to that in **9**. The structure of **9Pd** has been revealed by X-ray analysis, displaying a slightly curved structure with a large MPD (0.36 Å). Pd1 is coordinated with the three pyrrolic nitrogen atoms and one α-carbon of the N-confused pyrrole with a bond length of 2.034(6) Å for Pd1-N1, 2.029(5) Å for Pd1-N2, 1.956(5) Å for Pd1-N3, and 2.073(6) Å for Pd1-C. Pd2 is coordinated with the three pyrrolic nitrogen atoms and one oxygen atom with a bond length of 1.967(5) Å for Pd2-N4, 1.962(5) Å for Pd2-N5, 1.968(5) Å for Pd2-N6, and 2.002(5) Å for Pd2-O1.

The absorption spectra of **8**, **9**, **9Pd**, **9Ni**, and **9Cu** are shown in Fig. 7a. Characteristically, all these DNCRC [24]hexaphyrin complexes show ill-defined Soret bands and weak bands in the range of 800–1400 nm in line with their antiaromatic properties. Judging from the most red-shifted bands, the optical HOMO–LUMO gaps have been estimated to be 1.05, 1.15, 1.12, 1.05, and 1.08 eV for **8**, **9**, **9Pd**, **9Ni**, and **9Cu**, respectively.

Cyclic voltammetry (CV) and differential-pulse voltammetry (DPV) experiments were conducted and the redox potentials are summarized in Table 1.

The electrochemical properties were investigated by cyclic vol-tammetry and differential pulse voltammetry. Complex **8** showed reversible oxidation waves at 0.65 and 0.18 V and reduction waves at −1.05, −1.52, and −1.79 V, indicating the HOMO–LUMO gap to be 1.23 eV. Complexes **9**, **9Pd**, **9Ni**, and **9Cu** showed HOMO–LUMO gaps of 1.12, 1.19, 1.20, and 1.15 eV, respectively, roughly matching with the optical HOMO-LUMO gaps.

Further fabrications of **8** were attempted. Monobromide **8a** was obtained in 78% yield by treating **8** with 1 equiv. NBS at 0 °C (Fig. 8). The parent ion peak of **8a** was observed at m/z = 1196.1615 (calcd for $(C_{64}H_{53}BrN_6Pd_2)^+$ = 1196.1601 ($[M]^+$)) and its $^1H$ NMR spectrum showed nine peaks at 5.23–4.05 ppm, indicating an asymmetric substitution. Subsequently, directly β-to-β linked DNCRC [24]hexaphyrin dimer **10** was successfully obtained by reductive coupling of **8a** with $Ni(cod)_2$ in 61% yield. The parent ion peak of **10** was observed at m/z = 2234.4843 (calcd for $(C_{128}H_{106}N_{12}Pd_4)^+$ = 2234.4862 ($[M]^+$)). The $^1H$ NMR spectrum of **10** exhibited three singlets at 4.30, 3.91, and 3.74 ppm and three pairs of doublets in the range of 5.13–4.69 ppm. The structure of **10** has been confirmed by X-ray analysis as shown in Fig. 9. The dihedral angle between the two hexaphyrins is 54.2° and the bond length of the connecting C–C is 1.48 (1) Å. The first oxidation potentials of **10** were split at 0.31 and 0.15 V, indicating the electronic interaction between the two DNCRC [24]hexaphyrin units.

Dibromide **8b** was obtained in 86% yield by treating **8** with 2 equiv. NBS at 0 °C. The parent ion peak of **8b** was observed at m/z = 1274.0660 (calcd for $(C_{64}H_{52}Br_2N_6Pd_2)^+$ = 1274.0701 ($[M]^+$)) and its $^1H$ NMR spectrum showed two doublets at 5.22 and 4.84 ppm ($J$ = 4.7 Hz) and two singlets at 5.20 and 4.23 ppm, indicating a symmetric substitution. The single-crystal structure of **8b** was also determined by X-ray diffraction analysis, which shows its cen-trosymmetric dibromide feature. Finally, the Suzuki–Miyaura

**Fig. 5 | Synthesis of compounds 6, 7, 8, 9, 9Pd, 9Ni and 9Cu.** Reaction conditions: **a** X-phos, $Cs_2CO_3$, CsF, $Pd_2(dba)_3$, toluene, DMF, reflux, 48 h; **b** $M(OAc)_2$, NaOAc, $CHCl_3$, MeOH, reflux.

coupling reaction between **8b** and 5-boryl-10,15,20-triarylporphyrinato Ni^II **11** gave Ni^II porphyrin-DNCRC [24hexaphyrin-Ni^II porphyrin triad **12**, in 43% yield. The parent ion peak of **12** was observed at $m/z = 2975.2621$ (calcd for $(C_{188}H_{194}N_{14}Ni_2Pd_2)^+ = 2975.2429$ $([M]^+)$). The ^1H NMR spectrum showed two singlets at 5.94 and 5.58 ppm and a pair of doublets at 5.19 and 4.70 ppm, revealing its symmetric structure. The dihedral angle between the hexaphyrin and Ni^II porphyrin is 58.9° and the newly formed C−C bonds are 1.47 (1) Å long.

The absorption spectra of **8**, **10**, and **12** are shown in Fig. 7b. Dimer **10** shows bands at 395, 503, 635, 712, and 796 nm along with a weak tail. The three bands observed at longer wavelengths are considerably red-shifted in comparison to those of **8**. Triad **12** displays a Soret band of the Ni^II porphyrins at 416 nm. A red-shifted band of the DNCRC [24] hexaphyrin observed at 792 nm suggests its substantial electronic interaction with the Ni^II porphyrin.

In summary, 1,14-dibromotripyrrin **3b** underwent a self-coupling reaction under Pd-catalyzed Suzuki–Miyaura coupling conditions, giving DNCRC [24]hexaphyrin(1.1.0.1.1.0) Pd^II complexes **8** and **9**, along with 1,14-dibromotripyrrin–benzalacetone coupled products **6** and **7**. While triphyrin **6** is an aromatic [18] triphyrin(5.1.1), DNCRC [24]hexaphyrins **8** and **9** are stable antiaromatic expanded NCP showing paratropic ring currents, ill-defined absorption spectra with red-shifted weak absorption bands, and small HOMO−LUMO gaps. Monobromide **8a** was reductively dimerized to give **10** and dibromide **8b** was coupled with **11** to give Ni^II porphyrin-DNCRC [24hexaphyrin-Ni^II porphyrin triad **12**.

## Methods
### Materials and characterization
^1H NMR spectra (500 MHz) were taken on a Bruker ADVANCE-500 spectrometer, and chemical shifts were reported as the delta scale in ppm relative to $CHCl_3$ ($\delta = 7.260$ ppm) as an internal reference. UV/Vis absorption spectra were recorded on a Shimadzu UV-3600

spectrometer. MALDI-TOF mass spectra were obtained with a Bruker ultrafleXtreme MALDI-TOF/TOF spectrometer with *trans*-2-[3-(4-*tert*-Butylphenyl)-2-methyl-2-propenylidene]malononitrile (DCTB) as a matrix. X-ray data were taken on an Agilent Supernova X-ray diffractometer equipped with a large area CCD detector. Redox potentials were measured by cyclic voltammetry on a CHI900 scanning electrochemical microscope. Toluene and THF were distilled after refluxing with Na and benzophenone ketyl indicator under an argon atmosphere and stored over 3 Å molecular sieves in a glovebox for at least 12 h prior to use. DMF was refluxed with $CaH_2$ under an argon atmosphere for at least 3 h and distilled prior to use. Unless otherwise noted, materials obtained from commercial suppliers were used without further purification.

### Synthesis of 5 and 6
A suspension of 1,14-diboryltripyrrane **3a** (214.0 mg, 0.30 mmol), 2,4-dibromo-1-methoxybenzene (53.2 mg, 0.20 mmol), $Pd_2(dba)_3$ (18.3 mg, 0.02 mmol), X-Phos (38.1 mg, 0.08 mmol), $Cs_2CO_3$ (130.3 mg, 0.40 mmol), and CsF (60.7 mg, 0.40 mmol) in toluene/DMF suspension (4 mL/2 mL) was degassed through three freeze–pump–thaw cycles, and the reaction flask was purged with argon. The resulting mixture was refluxed for 48 h. The reaction mixture was diluted with $CHCl_3$. The organic layer was separated and washed with water, and dried over anhydrous sodium sulfate. Evaporation of the solvent followed by silica-gel column chromatography (eluent: $CH_2Cl_2/n$-hexane = 1:4, v/v) and recrystallization with $CH_2Cl_2$/MeOH gave **5** as green solids (5.6 mg, 0.010 mmol, 5% yield) and **6** as dark green solids (1.4 mg, 0.0020 mmol, 1% yield). **5**: ^1H NMR (500 MHz, $CDCl_3$) $\delta = 17.85$ (br, 1H, N-H), 12.50 (d, 1H, $J = 2.0$ Hz, Ph-H), 7.36 (dd, 1H, Ph-H), 6.87 (s, 4H, Mes-*m*-H), 6.69 (d, 1H, $J = 4.5$ Hz, β-H), 6.66 (d, 1H, $J = 4.5$ Hz, Ph-H), 6.58 (d, 1H, $J = 4.5$ Hz, β-H), 6.48 (d, 1H, $J = 4.5$ Hz, β-H), 6.46 (d, 1H, $J = 4.5$ Hz, β-H), 5.57 (m, 2H, β-H), 2.87 (s, 3H, OMe-H), 2.30 (s, 6H, Me-H) and 2.16 (s, 12H, Me-H) ppm. HR-MS (MALDI-TOF-MS): $m/z = 561.2794$, calcd for $(C_{39}H_{35}N_3O)^+ = 561.2775$ $([M]^+)$.

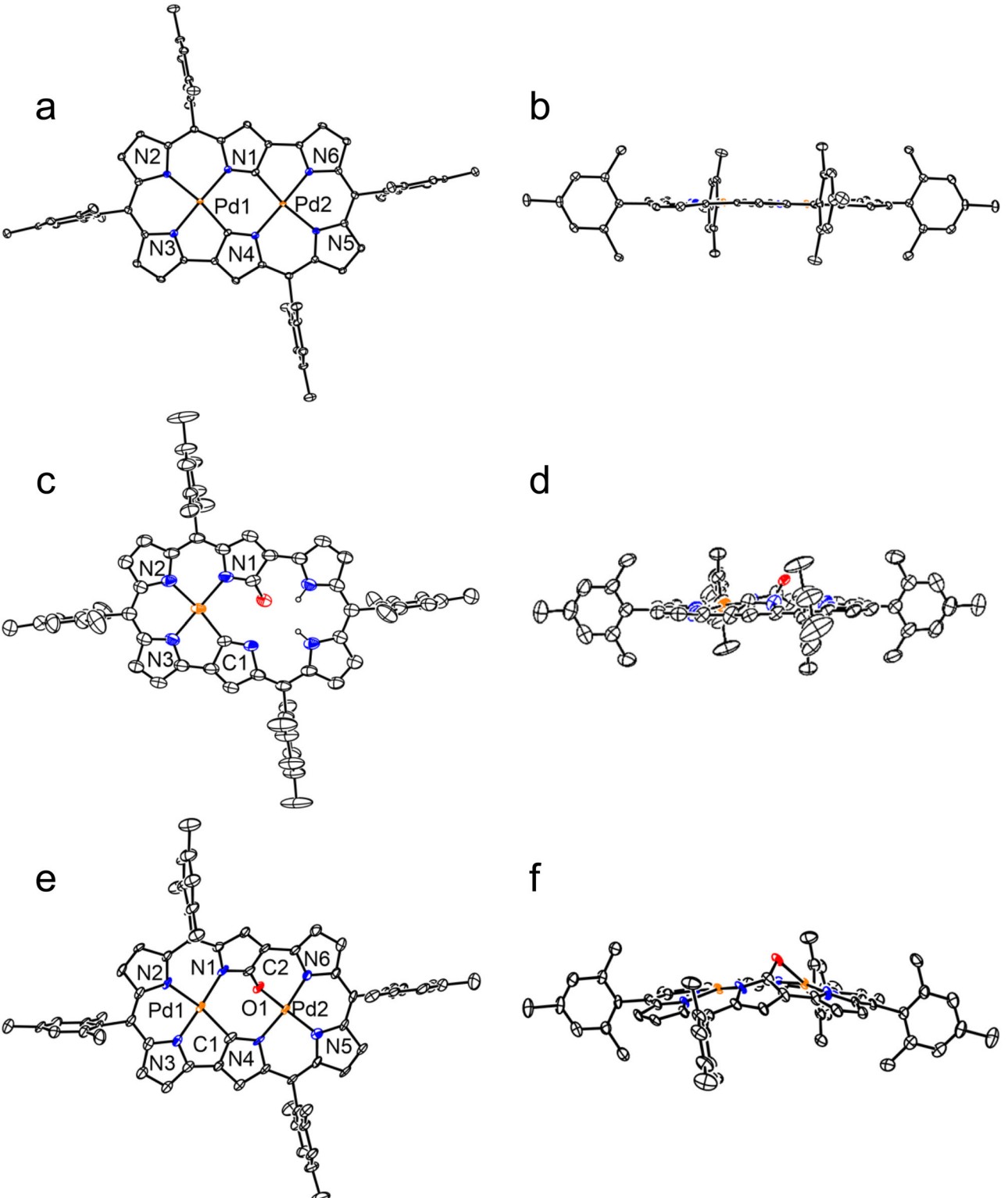

**Fig. 6 | X-ray crystal structures of 8, 9 and 9Pd. a** Top view of **8**, **b** side view of **8**, **c** top view of **9**, **d** side view of **9**, **e** top view of **9Pd**, **f** side view of **9Pd**. Ellipsoids are drawn at the 30% probability level. All hydrogen atoms except those connected to N atoms are omitted for clarity.

**6**: $^1H$ NMR (500 MHz, CDCl$_3$) $\delta = 9.74$ (s, 2H, vinyl-H), 8.36 (d, $J = 4.0$ Hz, 2H, $\beta$-H), 8.31 (s, 2H, $\beta$-H), 8.20 (d, $J = 7.2$ Hz, 4H, Ph-H), 8.08 (dd, $J = 4.0$, 1.5 Hz, 2H, $\beta$-H), 7.73 (t, 4H, Ph-H), 7.67 (t, $J = 6.0$ Hz, 2H, Ph-H), 7.28 (s, 4H, Mes-$m$-H), 3.87 (s, 2H, N-H), 2.63 (br, 6H, Me-H), 1.89 (br, 12H, Me-H) and $-4.13$ (s, 1H, N-H) ppm. $^{13}C$ NMR (126 MHz, CDCl$_3$) $\delta = 166.7$, 145.4, 139.8, 137.8, 137.6, 137.1, 134.8, 134.6, 132.7, 132.6, 128.2, 128.0, 127.9, 127.5, 127.1, 123.8, 123.4, 122.0, 106.8, 21.6 and 21.1 ppm.

UV/Vis (CH$_2$Cl$_2$): $\lambda_{max}$ ($\varepsilon$ [M$^{-1}$ cm$^{-1}$]) = 419 (21,000), 444 (25,000), and 652(6300) nm. HR-MS (MALDI-TOF-MS): $m/z = 689.3441$, calcd for (C$_{49}$H$_{43}$N$_3$O)$^+$ = 689.3401 ([$M$]$^+$).

### Synthesis of 6, 7, 8 and 9

A suspension of 1,14-dibromotripyrrin (100.0 mg, 0.16 mmol), Pd$_2$(dba)$_3$ (50.0 mg, 0.05 mmol), X-Phos (38.1 mg, 0.08 mmol), Cs$_2$CO$_3$

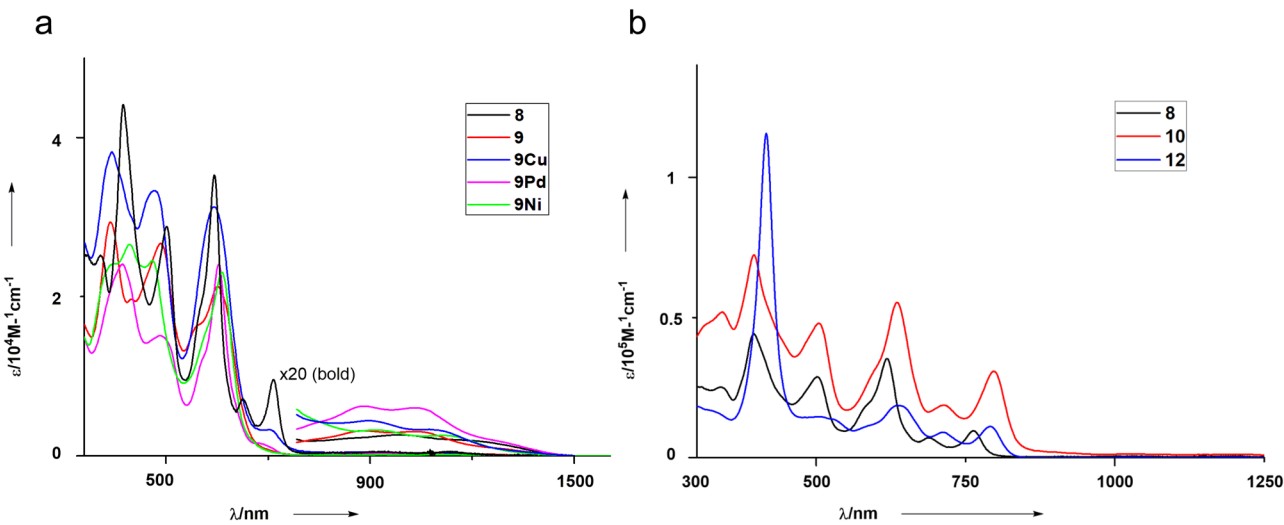

**Fig. 7 | Absorption spectra of 8, 9, 9Cu, 9Pd, 9Ni, 10 and 12 in CH₂Cl₂. a** UV–Vis absorption spectra of **8**, **9**, **9Pd**, **9Ni** and **9Cu** in CH₂Cl₂; **b** UV–Vis absorption spectra of **8**, **10** and **12** in CH₂Cl₂.

(130.3 mg, 0.40 mmol), and CsF (60.7 mg, 0.40 mmol) in toluene/DMF (4 mL/2 mL) was degassed through three freeze–pump–thaw cycles, and the reaction flask was purged with argon. The resulting mixture was stirred at reflux for 48 h. The reaction mixture was diluted with CHCl₃. The organic layer was separated and washed with water, and dried over anhydrous sodium sulfate. Evaporation of the solvent followed by silica-gel column chromatography (eluent: CH₂Cl₂/$n$-hexane = 1:3, v/v) and recrystallization with CH₂Cl₂/MeOH gave **6** as dark green solids (1.10 mg, 0.0016 mmol, 1% yield), **7** as blue solids (2.4 mg, 0.003 mmol, 2% yield), **8** as gray-green solids (8.4 mg, 0.0075 mmol, 9% yield), and **9** as emerald green solids (5.2 mg, 0.005 mmol, 10% yield).

**7:** ¹H NMR (500 MHz, CDCl₃) $\delta$ = 17.17 (br, 1H, OH), 7.29 (d, $J$ = 7.5 Hz, 2H, Ph-$o$-H), 7.13-7.08 (m, 5H, Ph-H), 7.04 (t, $J$ = 7.1 Hz, 1H, Ph-$p$-H), 6.99 (d, 2H, $J$ = 7.5 Hz, Ph-$o$-H), 6.91 (s, 1H, Mes-$m$-H), 6.89 (s, 1H, Mes-$m$-H), 6.88 (s, 1H, Mes-$m$-H), 6.87 (s, 1H, Mes-$m$-H), 6.60−6.59 (m, 2H, $\beta$-H), 6.51 (d, $J$ = 4.5 Hz, 1H, $\beta$-H), 6.26 (d, $J$ = 4.3 Hz, 1H, $\beta$-H), 6.24 (d, $J$ = 5.0 Hz, 1H, $\beta$-H), 5.91 (d, $J$ = 4.3 Hz, 1H, $\beta$-H), 5.68 (d, $J$ = 3.0 Hz, 1H, $sp^3$C-H), 4.23 (d, $J$ = 7.7 Hz, 1H, $sp^3$C-H), 3.25 (dd, $J$ = 16.5, 7.8 Hz, 1H, $sp^3$C-H), 2.63 (dd, $J$ = 16.5, 3.1 Hz, 1H, $sp^3$C-H), 2.34 (s, 3H, Me-H), 2.32 (s, 3H, Me-H), 2.18 (s, 3H, Me-H), 2.09 (s, 3H, Me-H), 2.05 (s, 3H, Me-H) and 1.96 (s, 3H, Me-H) ppm. ¹³C NMR (126 MHz, CDCl₃) $\delta$ = 179.7, 179.0, 146.9, 144.6, 141.2, 141.0, 139.7, 139.5, 137.7, 137.6, 137.54, 137.47, 137.0, 136.9, 134.2, 133.8, 133.6, 131.7, 130.0, 128.51, 128.46, 128.2, 128.1, 128.0, 127.8, 126.8, 126.4, 122.2, 121.9, 118.7, 64.4, 54.6, 54.2, 45.9, 21.3, 21.2,

20.7, 20.4(2C) and 20.2 ppm. (The absence of some peaks in ¹³C NMR may be due to overlapping.) UV/Vis (CH₂Cl₂): $\lambda_{max}$ ($\varepsilon$ [M⁻¹ cm⁻¹]) = 355 (21,100), 400 (26,500), and 667 (6600) nm. HR-MS (MALDI-TOF-MS): $m/z$ = 811.2408, calcd for (C₄₉H₄₃N₃O₂Pd)⁺ = 811.2402 ([$M$]⁺).

**8:** ¹H NMR (500 MHz, CDCl₃) $\delta$ = 6.72 (s, 4H, Mes-$m$-H), 6.68 (s, 4H, Mes-$m$-H), 5.20 (d, $J$ = 4.5 Hz, 2H, $\beta$-H), 5.18 (d, $J$ = 4.5 Hz, 2H, $\beta$-H), 5.12 (d, $J$ = 4.5 Hz, 2H, $\beta$-H), 4.80 (d, $J$ = 4.5 Hz, 2H, $\beta$-H), 4.02 (s, 2H, $\beta$-H), 2.25 (s, 12H, Me-H), and 2.18−2.16 (m, 24H, Me-H). ¹³C NMR (126 MHz, CDCl₃) $\delta$ = 223.8, 161.5, 153.1, 152.0, 149.5, 148.3, 143.9, 138.5, 137.3, 137.1, 136.1, 132.5, 131.9, 131.8, 131.6, 128.5, 127.9, 127.8, 121.2, 120.4, 114.3, 29.8, 21.0, 19.9, and 19.4 ppm. (The absence of some peaks in ¹³C NMR may be due to overlapping). UV/Vis (CH₂Cl₂): $\lambda_{max}$ ($\varepsilon$ [M⁻¹ cm⁻¹]) = 340 (25,200), 396 (44,100), 502 (28,800), 618 (35,300), 690 (7100), 763 (7600), and 1082 (100) nm. HR-MS (MALDI-TOF-MS): $m/z$ = 1120.2568, calcd for (C₆₄H₅₄N₆Pd₂)⁺ = 1120.2504 ([$M$]⁺).

**9:** ¹H NMR (500 MHz, CDCl₃) $\delta$ = 21.89 (s, 1H, N-H), 19.85 (s, 1H, N-H), 6.68-6.64 (m, 8H, Mes-$m$-H), 5.22 (d, $J$ = 4.5 Hz, 1H, $\beta$-H), 5.13 (d, $J$ = 4.5 Hz, 1H, $\beta$-H), 5.10 (dd, $J$ = 4.5, 2.0 Hz, 1H, $\beta$-H), 5.07 (dd, $J$ = 4.5, 2.0 Hz, 1H, $\beta$-H), 5.02 (d, $J$ = 4.5 Hz, 1H, $\beta$-H), 4.88 (d, $J$ = 4.5 Hz, 1H, $\beta$-H), 4.73 (d, $J$ = 4.5 Hz, 1H, $\beta$-H), 4.62 (d, $J$ = 4.5 Hz, 1H, $\beta$-H), 4.30 (s, 1H, $\beta$-H), 3.79 (s,1H, $\beta$-H) and 2.22-2.12 (m, 36H, Me-H). ¹³C NMR (126 MHz, CDCl₃) $\delta$ = 215.8, 187.1, 166.2, 157.6, 155.8, 155.4, 151.3, 149.7, 140.3, 137.64, 137.56, 137.1, 137.0, 135.9, 135.7, 132.8, 132.1, 131.9, 131.4, 130.7, 128.4, 128.0, 127.9, 126.7, 125.7, 124.4, 123.8, 123.0, 120.3, 116.0, 113.2, 110.7, 100.0, 29.7, 21.0, 20.0, 19.5, and 19.1 ppm. (The absence of some peaks in ¹³C NMR may be due to overlapping.) UV/Vis (CH₂Cl₂): $\lambda_{max}$ ($\varepsilon$ [M⁻¹ cm⁻¹]) = 364 (29,400), 487 (26,700), 628 (21,200), 987 (200), and 1119 (200) nm. HR-MS (MALDI-TOF-MS): $m/z$ = 1030.3687, calcd for (C₆₄H₅₆N₆OPd)⁺ = 1030.3566 ([$M$]⁺).

**Synthesis of 8a**

A solution of NBS (3.2 mg, 0.018 mmol) in CHCl₃ (10 mL) was added to a solution of **8** (20.0 mg, 0.018 mmol) in CHCl₃ (10 mL) dropwise at 0 °C. After the consumption of **8** was confirmed by TLC monitoring, the reaction mixture was diluted with CHCl₃. The organic layer was separated and washed with water, and dried over anhydrous sodium sulfate, evaporation of the solvent was followed by silica-gel column chromatography (eluent: CH₂Cl₂/$n$-hexane = 1:4, v/v) and recrystallization with CH₂Cl₂/MeOH. **8a** was obtained as grass-green solids (16.7 mg, 0.014 mmol, 78% yield).

**Table 1 | Electrochemical properties of 8, 9, 9Pd, 9Ni, and 9Cu in CH₂Cl₂ with 0.1 M $n$-Bu₄NPF₆ᵃ**

| Compound | $E_{ox.3}$ (V) | $E_{ox.2}$ (V) | $E_{ox.1}$ (V) | $E_{red.1}$ (V) | $E_{red.2}$ (V) | $E_{red.3}$ (V) | $E_{red.4}$ (V) | $\Delta E_{HL}$ (eV) |
|---|---|---|---|---|---|---|---|---|
| **8** | | 0.65 | 0.18 | −1.05 | −1.52 | −1.79 | | 1.23 |
| **9** | | 0.63 | 0.18 | −0.94 | −1.52 | | | 1.12 |
| **9Pd** | | 0.68 | 0.21 | −0.98 | | | | 1.19 |
| **9Ni** | | 0.67 | 0.20 | −1.00 | −1.48 | | | 1.20 |
| **9Cu** | | 0.62 | 0.15 | −1.00 | −1.50 | | | 1.15 |
| **10** | 0.75 | 0.31 | 0.15 | −1.06 | −1.33 | −1.36 | −1.78 | 1.21 |
| **12** | | 0.68 | 0.24 | −1.08 | −1.74 | | | 1.32 |

Cyclic voltammetry experiments indicated that most of the redox processes were reversible (for details, see Supplementary Information). Working electrode: glassy carbon, counter electrode: Pt wire, reference electrode: Ag/AgCl.
ᵃPotentials (V) were determined vs. ferrocene/ferrocenium ion by differential pulse voltammetry.

**Fig. 8 | Synthesis of compounds 8a, 8b, 10 and 12.** Reaction conditions: **a** NBS (1.0 equiv or 2.0 equiv.), 0 °C, CHCl₃; **b** Ni(cod)2 (1.0 equiv), 2,2'-bipyridine (1.0 equiv), reflux, 24 h; **c** SPhos Pd G2 (10 mol %), K₃PO₄ (5.0 equiv.), THF/H₂O = 20:1, reflux, 36 h.

**8a:** $^1$H NMR (500 MHz, CDCl₃) $\delta$ = 6.71 (s, 4H, Mes-$m$-H), 6.67 (s, 4H, Mes-$m$-H), 5.23 (d, $J$ = 4.5 Hz, 1H, $\beta$-H), 5.21 (d, $J$ = 4.5 Hz, 1H, $\beta$-H), 5.15 (s, 1H, $\beta$-H), 5.14 (s, 1H, $\beta$-H), 5.13 (s, 1H, $\beta$-H), 4.81-4.80 (m, 2H, $\beta$-H), 4.17 (s, 1H, $\beta$-H), 4.05 (s, 1H, $\beta$-H), 2.22 (br, 6H, Me-H), 2.21 (br, 6H, Me-H), and 2.18-2.15 (m 24H, Me-H). HR-MS (MALDI-TOF-MS): $m/z$ = 1196.1615, calcd for (C₆₄H₅₃BrN₆Pd₂)⁺ = 1196.1601 ([$M$]⁺).

### Synthesis of 8b

A solution of NBS (6.4 mg, 0.036 mmol) CHCl₃ (10 mL) was added to a solution of **8** (20.0 mg, 0.018 mmol) in CHCl₃ (20 mL) dropwise at 0 °C. After the consumption of **8** and **8a** was confirmed by TLC monitoring, the reaction mixture was diluted with CHCl₃. The organic layer was separated and washed with water, and dried over anhydrous sodium

sulfate, Evaporation of the solvent was followed by silica-gel column chromatography (eluent: CH₂Cl₂/$n$-hexane = 1:4, v/v) and recrystallization with CH₂Cl₂/MeOH gave **8b** as grass green solids (19.7 mg, 0.016 mmol, 86% yield).

**8b:** $^1$H NMR (500 MHz, CDCl₃) $\delta$ = 6.72 (s, 4H, Mes-$m$-H), 6.68 (s, 4H, Mes-$m$-H), 5.21 (d, $J$ = 4.5 Hz, 2H, $\beta$-H), 5.20 (s, 2H, $\beta$-H), 4.84 (d, $J$ = 4.5 Hz, 2H, $\beta$-H), 4.22 (s, 2H, $\beta$-H), 2.22 (br, 12H, Me-H), 2.18 (br, 6H, Me-H), and 2.16 (br, 18H, Me-H). HR-MS (MALDI-TOF-MS): $m/z$ = 1274.0660, calcd for (C₆₄H₅₂Br₂N₆Pd₂)⁺ = 1274.0701 ([$M$]⁺).

### Synthesis of 9Cu

A suspension of **9** (20.0 mg, 0.019 mmol), Cu(OAc)₂ (37.8 mg, 0.19 mmol), and NaOAc (15.6 mg, 0.19 mmol) in CHCl₃/MeOH

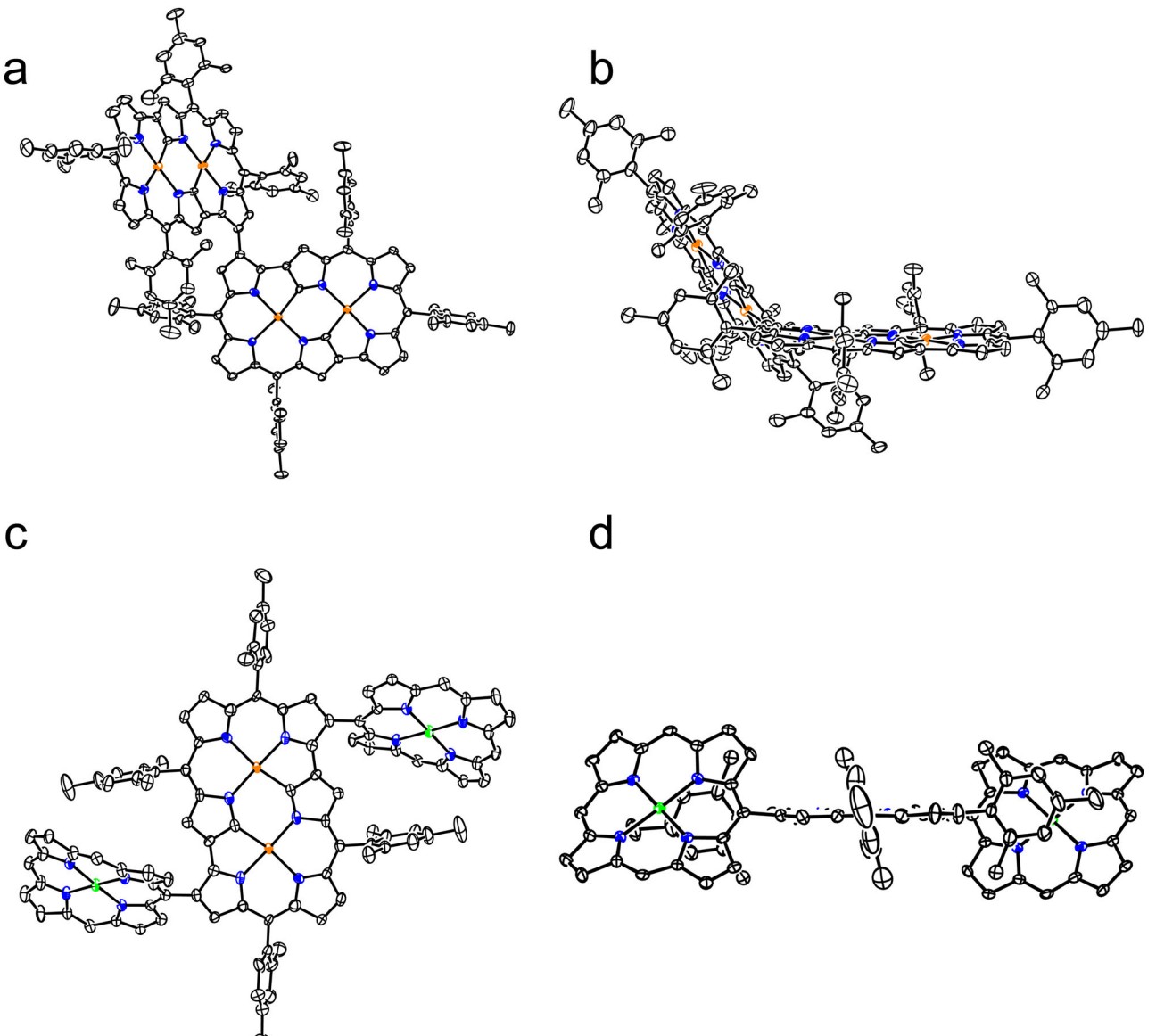

**Fig. 9 | X-ray crystal structures of 10 and 12. a** Top view of **10**, **b** side view of **10**, **c** top view of **12**, and **d** side view of **12**. Ellipsoids are drawn at the 30% probability level. All hydrogen atoms are omitted for clarity. *Meso*-substituents of porphyrin parts are omitted for clarity. Nickel atom, green.

(20 mL/10 mL) was stirred at 65 °C overnight. The solvent was removed under reduced pressure. The product was separated by silica-gel column chromatography (eluent:$CH_2Cl_2$/$n$-hexane = 1:1, v/v) and recrystallization with $CH_2Cl_2$/MeOH gave **9Cu** as brown crystals (17.6 mg, 0.016 mmol, 85% yield). UV/Vis ($CH_2Cl_2$): $\lambda_{max}$ ($\varepsilon$ [$M^{-1} cm^{-1}$]) = 368 (38,200), 472 (33,300), 630 (24,000), 994 (300) and 1166 (200) nm. HR-MS (MALDI-TOF-MS): $m/z$ = 1091.2730, calcd for $(C_{64}H_{54}CuN_6OPd)^+$ = 1091.2702([$M + H$]$^+$).

**Synthesis of 9Pd and 9Ni**

By following the method used for the synthesis of **9Cu**−**9Pd** and **9Ni** were synthesized upon treatment with Pd(OAc)$_2$ and Ni(OAc)$_2$·4H$_2$O in 80% and 95% yields. **9Pd**: $^1$H NMR (500 MHz, CDCl$_3$) $\delta$ = 6.90 (br, 1H, Mes-$m$-H), 6.76 (br, 1H, Mes-$m$-H), 6.73 (br, 1H, Mes-$m$-H), 6.72 (br, 2H, Mes-$m$-H), 6.69 (br, 1H, Mes-$m$-H), 6.65 (br, 2H, Mes-$m$-H), 5.40 (d, $J$ = 4.6 Hz, 1H, $\beta$-H), 5.38−5.33 (m, 4H, $\beta$-H), 5.10 (d, $J$ = 4.6 Hz, 1H, $\beta$-H), 5.00 (d, $J$ = 4.6 Hz, 1H, $\beta$-H), 4.86−4.83 (m, 2H, $\beta$-H), 4.09 (s, 1H, $\beta$-H), 2.85 (s, 3H, Me-H), 2.37 (s, 3H, Me-H), 2.31 (s, 3H, Me-H), 2.22 (s, 3H, Me-H), 2.20 (s, 3H, Me-H), 2.19 (s, 9H, Me-H), 2.16 (s, 3H, Me-H), 2.02 (s, 3H, Me-H), 1.99 (s, 3H, Me-H), and 1.65 (s, 3H, Me-H). $^{13}$C NMR (126 MHz,

CDCl$_3$) $\delta$ = 212.4, 184.4, 159.0, 158.0, 154.0, 150.4, 150.2, 150.0, 147.7, 145.9, 142.8, 141.7, 137.7, 137.4, 137.3, 137.2, 136.8, 136.5, 136.3, 136.3, 136.1, 136.1, 136.0, 136.0, 132.2, 131.6, 131.2, 131.2, 129.9, 129.9, 129.3, 128.2, 128.0, 127.8, 127.8, 127.7, 124.5, 121.5, 120.3, 118.0, 116.8, 113.2, 21.0, 21.0, 20.8, 20.2, 19.7, 19.6 and 19.4 ppm. (The absence of some peaks in $^{13}$C NMR may be due to overlapping.) UV/Vis ($CH_2Cl_2$): $\lambda_{max}$ ($\varepsilon$ [$M^{-1} cm^{-1}$]) = 411 (26,600), 316 (5700) and 490 (38,000) nm. HR-MS (MALDI-TOF-MS): $m/z$ = 1136.2426, calcd for $(C_{64}H_{54}N_6OPd_2)^+$ = 1136.2453 ([$M + H$]$^+$).

**9Ni**: $^1$H NMR (500 MHz, CDCl$_3$) $\delta$ = 6.89 (br, 1H, Mes-$m$-H), 6.70 (br, 1H, Mes-$m$-H), 6.68 (br, 1H, Mes-$m$-H), 6.68 (br, 1H, Mes-$m$-H), 6.65 (br, 1H, Mes-$m$-H), 6.63 (br, 1H, Mes-$m$-H), 6.60 (br, 2H, Mes-$m$-H), 5.20 (d, 1H, $J$ = 4.6 Hz, $\beta$-H), 5.14-5.12 (m, 2H, $\beta$-H), 5.10 (d, 1H, $J$ = 4.6 Hz, $\beta$-H), 5.07 (d, 1H, $J$ = 4.6 Hz, $\beta$-H), 4.90 (d, 1H, $J$ = 4.6 Hz, $\beta$-H), 4.83 (d, 1H, $J$ = 4.6 Hz, $\beta$-H), 4.61-4.60 (m, 2H, $\beta$-H), 3.77 (s, $\beta$-H), 2.95 (s, 3H, Me-H), 2.23 (s, 3H, Me-H), 2.22 (s, 3H, Me-H), 2.20 (s, 3H, Me-H), 2.20 (s, 3H, Me-H), 2.18 (s, 3H, Me-H), 2.16 (s, 3H, Me-H), 2.15 (s, 3H, Me-H), 2.12 (s, 3H, Me-H), 2.10 (s, 3H, Me-H), 2.08 (s, 3H, Me-H) and 1.62 (s, 3H, Me-H). $^{13}$C NMR (126 MHz, CDCl3) $\delta$ = 217.9, 187.1, 160.2, 158.7, 154.42, 154.39, 152.30, 152.0, 151.3, 147.6, 146.6, 144.1, 143.39, 139.8, 137.5, 137.3, 137.2, 136.0, 135.9, 132.0, 131.6, 131.4, 131.0, 130.2, 128.5, 128.2, 128.0, 127.9,

127.6, 122.6, 121.1, 119.9, 118.5, 113.2, 29.7, 21.0, 20.9, 20.8, 20.0, 19.8, 19.5, 19.4, 19.3 and 19.2 ppm. (The absence of some peaks in $^{13}C$ NMR may be due to overlapping.) UV/Vis ($CH_2Cl_2$): $\lambda_{max}$ ($\varepsilon$ [$M^{-1}$ $cm^{-1}$]) = 393 (19,000), 836 (4000) and 931 (4600) nm. HR-MS (MALDI-TOF-MS): $m/z$ = 1086.2635, calcd for ($C_{64}H_{54}NiN_6OPd$)$^+$ = 1086.2758 ([$M$]$^+$).

## Synthesis of 10

A THF solution (5 mL) of **8a** (20.0 mg, 0.017 mmol), Ni(cod)$_2$ (4.6 mg, 0.017 mmol), and 2,2′-bipyridine (2.7 mg, 0.017 mmol) in a 50 mL Schlenk tube was purged with argon. The mixture was refluxed for 24 h. The reaction mixture was diluted with CHCl$_3$. The organic layer was separated and washed with water, and dried over anhydrous sodium sulfate. Evaporation of the solvent followed by silica-gel column chromatography (eluent: $CH_2Cl_2$/$n$-hexane = 1:3, v/v) and recrystallization with $CH_2Cl_2$/MeOH gave **10** as green solids (11.6 mg, 0.005 mmol, 61% yield).

**10**: $^1H$ NMR (500 MHz, CDCl$_3$) $\delta$ = 6.81 (s, 2H, Mes-$m$-H), 6.70-6.69 (m, 8H, Mes-$m$-H), 6.62 (s, 2H, Mes-$m$-H), 6.59 (s, 2H, Mes-$m$-H), 5.13 (d, 2H, $J$ = 4.5 Hz, $\beta$-H), 5.12 (d, 2H, $J$ = 4.5 Hz, $\beta$-H), 5.08 (d, 2H, $J$ = 4.5 Hz, $\beta$-H), 4.95 (d, 2H, $J$ = 4.0 Hz, $\beta$-H), 4.70 (d, 2H, $J$ = 4.5 Hz, $\beta$-H), 4.69 (d, 2H, $J$ = 4.0 Hz, $\beta$-H), 4.30 (s, 2H, $\beta$-H), 3.91 (s, 2H, $\beta$-H), 3.74 (s, 2H, $\beta$-H), 2.59 (s, 6H, Me-H), 2.37 (s, 6H, Me-H), 2.23 (br, 18H, Me-H), 2.19 (br, 9H, Me-H), 2.17 (br, 9H, Me-H), 2.13 (br, 9H, Me-H), 2.12 (br, 9H, Me-H), 2.05 (s, 6H, Me-H) and 2.02 (s, 6H, Me-H). UV/Vis ($CH_2Cl_2$): $\lambda_{max}$ ($\varepsilon$ [$M^{-1}$ $cm^{-1}$]) = 395 (72600), 503 (48,400), 635 (55,200), 712 (18,700), and 796 (30,700) nm. HR-MS (MALDI-TOF-MS): $m/z$ = 2234.4843, calcd for ($C_{128}H_{106}N_{12}Pd_4$)$^+$ = 2234.4862 ([$M$]$^+$).

## Synthesis of 12

A THF−$H_2O$ suspension (2 mL/0.1 mL) of **8b** (20.0 mg, 0.016 mmol), **11** (34.0 mg, 0.032 mmol), Sphos Pd G2 (1.2 mg, 0.0016 mmol) and $K_3PO_4$ (6.5 mg, 0.027 mmol) in a 50 mL Schlenk tube was purged with argon. The mixture was refluxed for 36 h. The reaction mixture was diluted with CHCl$_3$. The organic layer was separated and washed with water, and dried over anhydrous sodium sulfate. Evaporation of the solvent followed by silica-gel column chromatography (eluent: $CH_2Cl_2$/$n$-hexane = 1:3, v/v) and recrystallization with $CH_2Cl_2$/MeOH gave **12** as dark green solids (20.5 mg, 0.007 mmol, 43% yield).

$^1H$ NMR (500 MHz, CDCl$_3$) $\delta$ = 8.89 (d, $J$ = 4.9 Hz, 4H, Por-$\beta$-H), 8.75 (dt, $J$ = 8.6, 4.9 Hz, 12H, Por-$\beta$-H), 7.76 (s, 4H, Por-Ar-H), 7.71 (s, 2H, Por-Ar-H), 6.74 (s, 4H, Mes-$m$-H), 5.94 (s, 2H, $\beta$-H), 5.58 (s, 2H, $\beta$-H), 5.19 (d, $J$ = 4.5 Hz, 2H, $\beta$-H), 4.70 (d, $J$ = 4.5 Hz, 2H, $\beta$-H), 2.45 (s, 12H, Me-H), 2.14 (s, 6H, Me-H), 1.70 (s, 18H, Me-H) and 1.60 (s, 108H, $t$-Bu-H). UV/Vis ($CH_2Cl_2$): $\lambda_{max}$ ($\varepsilon$ [$M^{-1}$ $cm^{-1}$]) = 416 (115,600), 636 (18,300), 714 (9300) and 792 (11,200) nm. HR-MS (MALDI-TOF-MS): $m/z$ = 2975.2621, calcd for ($C_{188}H_{194}N_{14}Ni_2Pd_2$)$^+$ = 2975.2429 ([$M$]$^+$).

## Data availability

The X-ray crystallographic coordinates for structures reported in this study have been deposited at the Cambridge Crystallographic Data Centre (CCDC), under deposition numbers 2239867, 2239868, 2239869, 2239872, 2239870, 2239871, 2278452, 2239873, 2239874 (**6, 7, 8, 8a, 8b, 9, 9Pd, 10, 12**). These data can be obtained free of charge from The Cambridge Crystallographic Data Centre via www.ccdc.cam. ac.uk/data_request/cif. Original $^1H$ NMR and $^{13}C$ NMR spectra, UV/vis absorption spectra, X-ray crystal data, electrochemical data, HR-MS Spectra, and DFT calculation results of the compounds generated in this study are provided in the Supplementary Information file.

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

## Acknowledgements

The work is supported by the National Natural Science Foundation of China (Grant Nos. 22071052 (J.S.), 21772036 (J.S.), 22271091 (L.X.), 22201072 (Y.R.)), Science and Technology Planning Project of Hunan

Province (2018TP1017 (J.S.)), and Science and Technology Innovation Program of Hunan Province (2021RC4059 (J.S.), 2022WZ1019 (M.Z.)).

## Author contributions

J.S. designed and conducted the project. F.L., L.L., and H.W. performed the synthesis and characterization and measured the optical and electrochemical properties. L.X., M.Z., and Y.R. performed X-ray diffraction analysis and DFT calculations. A.O. and J.S. prepared the manuscript.

## Competing interests

The authors declare no competing interests.
