## [Peer Review File · Nature Communications]

Doubly N-Confused and Ring-Contracted [24]Hexaphyrin Pd-Complexes as Stable Antiaromatic N-Confused Expanded PorphyrinsReviewers' Comments:

Reviewer #1:

Remarks to the Author:

[note from the Editor: Please see attached file]

In the present manuscript, the author introduced the synthesis of Doubly N-Confused and Ring-Contracted [24]Hexaphyrin Pd-Complexes and investigated the aromaticity using of the NICS method.

In general, aromatic compounds have high thermodynamic stability and are substantially more stable than antiaromatic compounds. Various indices of aromaticity exist, however as we know, the concept of aromaticity is complicated and a single value alone is often not enough to define a global property of the molecule.

There are many other reports that the NICS values calculated at the ring centers of polycyclic π systems do not always reflect the aromaticities of the rings. See, for example:

- 1) J. Aihara, Incorrect NICS-Based Prediction on the Aromaticity of the Pentalene Dication, *Bulletin of the Chemical Society of Japan*, 2004,77, 101-102.
- 2) S. Fias, FV Damme, P. Bultinck, Multidimensionality of Delocalization Indices and Nucleus Independent Chemical Shifts in Polycyclic Aromatic Hydrocarbons, *Journal of Computational Chemistry*, 2008, 29, 358-366.
- 3) S. Fias, FV Damme, P. Bultinck, Multidimensionality of Delocalization Indices and Nucleus-Independent Chemical Shifts in Polycyclic Aromatic Hydrocarbons II: Proof of Further No locality, *Journal of Computational Chemistry*, 2010, 31, 2286-2293.
- 4) P. Bultinck, S.Fias, R. Ponec, Local Aromaticity in Polycyclic Aromatic Hydrocarbons: Electron Delocalization versus Magnetic Indices, *Chemistry-A European Journal*, 2006, 12, 8813-8818.
- 5) J. Aihara, H. Kanno, Local Aromaticities in Large Polyacene Molecules, *Journal of Physical Chemistry A*, 2005, 109, 3717-3721.
- 6) J. Aihara, Nucleus-independent chemical shifts and local aromaticities in large polycyclic aromatic hydrocarbons, *Chemical Physics Letters*, 2002,365,34-39.
- 7) J. Aihara, Spherical aromaticity in fullerenes and the nucleus-independent chemical shifts at the cage centers, *Bulletin of the Chemical Society of Japan.*, 2003,76, 103-105.
- 8) J. Poater, I. Garca-Cruz, F. Illas, M. Sola, Discrepancy between common local aromaticity measures in a series of carbazole derivatives, *Physical Chemistry . Chemical Physics*, 2004 , 6 , 314-318.
- 9) A. Kerim, Aromaticity and kinetic stability of fullerene C₃₆ isomers and their molecular ions, *Journal of Molecular Modeling*, 2011,17,3257–3263.
- 10) J. Aihara, Re-interpretation of the 2(N+1)2 rule for polyhedral closed-shell π -systems, *Chemical Physics Letters*, 2003, 375,571-575.

Thus, NICS are very problematic for different reasons and such a comparison is not trustable.

The manuscript contains some scientific errors:

Page 6, line 123: “**8** and **9** are the first example of stable antiaromatic N-confused expanded porphyrins”.

I do not entirely agree with the statements.

In view of my critical comments listed above, my recommendation is to reject this manuscript for publication as an article in *Nature Communications*.

Ablikim Kerim

School of Chemical Engineering and Technology, Xinjiang University, Urumqi, 830046, Xinjiang,

People's Republic of China

Hossein Fallah-Bagher-Shaidaei, Chaitanya S. Wannere, Clémence Corminboeuf, Ralph Puchta, and Paul v. R. Schleyer, Which NICS Aromaticity Index for Planar π Rings Is Best? *Org. Lett.* 2006, 8, 5, 863–866, doi.org/10.1021/ol0529546

Reviewer #2:

Remarks to the Author:

The article describes the synthesis of several novel porphyrinoids from the palladium-mediated coupling of 1,14-dibromotripyrrin 3b. Homocoupling of 3b afforded doubly N-confused contracted hexaphyrin Pd(II) complexes, while hetero-coupling of 3b with dibenzylideneacetone yielded aromatic triphyrin 6. The products were fully characterized, and the X-ray diffraction analysis nicely elucidated their structures. The aromaticity and antiaromaticity of compounds 6 and 8 were investigated based on ¹H NMR analysis and DFT calculations. The authors obtained brominated hexaphyrins through regioselective bromination of 8, which were employed to synthesize dimer 10 and porphyrin-appended hexaphyrin 12.

Doubly N-confused contracted hexaphyrin Pd(II) complex 8 is more appealing than other molecules because 8 is the first example of stable antiaromatic N-confused expanded porphyrins. To focus on this intriguing novel hexaphyrin, the authors should eliminate Scheme 2. In addition, the choice of reagents in Scheme 3 seems unreasonable. To avoid the formation of undesirable by-products from dba, the authors should use palladium complexes without the dba ligand, such as Pd(PPh₃)₄ and allylPdCp, to improve the yield of compound 8. Moreover, the formation of 8 from dibromotripyrrin 3b is fascinating from the mechanistic viewpoint. Unfortunately, the possible reaction mechanism is not discussed in the present manuscript.

Regioselectivity in the bromination of compound 8 should also be discussed. The purpose of the synthesis of compounds 10 and 12 is not clear. The formation of these molecules is not very surprising, and the authors should demonstrate their unique properties originated from the intrinsic property of the contracted hexaphyrin core.

The manuscript would be much improved if the authors could find a better synthesis of the antiaromatic hexaphyrin palladium complex 8 and demonstrate its specific properties and functions.

Reviewer #3:

Remarks to the Author:

This is an interesting and insightful study on the synthesis of N-confused expanded porphyrins and an aromatic keto-triphyrin. Pd catalyzed cross-coupling of a diboryltripyrane (or tripyrrin?) with 2,4-dibromoanisole gave a contracted benzoporphyrin together with a tripyrrolic byproduct. At first glance it is hard to see where this comes from but the structure results from incorporation of dibenzalacetone from the catalyst Pd₂(dba)₂. The keto-triphyrin is aromatic due to polarization of the keto-moiety. The calculated NICS value is -12.15 ppm. This is an interesting system but the authors should note that related keto-carbaporphyrins have been reported (Inorg. Chem. 2017, 56, 11426; Org. Chem. Front. 2022, 9, 5440). In carbaporphyrin systems the aromatic circuit competes with antiaromatic pathways resulting in weakly aromatic structures. However, in a related chlorin, the antiaromatic pathway is no longer a possibility and as is the case here the structure becomes strongly aromatic. Initially, the authors obtained a low yield of the keto-triphyrin but a more targeted approach raised the yield to 8.8%. As a slight point of confusion, the authors refer to 3a as a tripyrrin but it is shown in Scheme 2 as a tripyrane. In Scheme 3, structure 3b is written as a tripyrrin, Is structure 3a a mistake? Pd catalyzed coupling of dibromotripyrrin 3b gave two hexaphyrin products together with the keto-triphyrin and an unusual tripyrrolic palladium complex. The byproducts both originated from incorporation of dibenzalacetone from the catalyst. Hexaphyrin 8 was isolated as a bis-Pd(II) complex and has two N-confused pyrrole units. The system essentially have two carbacorrole-like binding pockets. Product 9 is a similar "contracted" hexaphyrin but is a mono-palladium complex. In this case, one of the confused rings has been oxidized to give an internal keto-unit. Compound 9 can be further metalated to give bis-Pd, Pd-Ni and Pd-Cu complexes. This is an interesting series of metalated doubly-confused porphyrinoids and importantly the possibilities for metalation are considerably altered compared to the doubly N-confused hexaphyrins previously reported by Furuta. Bromination of the bis-Pd(II) complex, followed by coupling with Ni(cod)₂ gave a selectively linked dimer. In addition, dibromination, followed by Suzuki-Miyaura cross-coupling with a nickel(II) porphyrin gave a diporphyrinylhexaphyrin triad. Excellent yields are reported. The electronic absorption spectra indicate

that there are significant electronic interactions. In addition to spectroscopic and electrochemical characterization, X-ray crystal structures of 6, 7, 8, 9, 11 and 12 are provided. The only thing that seems to be missing is carbon-13 NMR. Is there a reason that carbon-13 NMR spectra are not included? While these will give limited information compared to proton NMR spectroscopy, I would have expected these to be included. If the samples are not sufficiently soluble to obtain this data, this should be noted.

Overall, this is an excellent study. I found a few minor errors but otherwise that is about it.

Page 1: "new comers" should be "newcomers", although after 30 years I am not sure NCPs can still be considered to be newcomers any more.

Line 104: "NMR spectrum" rather than just "NMR"

Line 108: a NICS value of 5.41 is somewhat borderline for antiaromaticity.

Line 131 and elsewhere: 5.40~4.09 ppm and 5.20~3.77 ppm should be 5.40-4.09 ppm and 5.20-3.77 ppm. These are ranges not approximations.

Experimental section: column chromatography was performed using CH₂Cl₂/n-hexane but the authors never provide the ratio of the solvents. Was it 1:1 or something else entirely? Whatever the proportions, and these are likely to vary from experiment to experiment, the ratios should be provided.

Reference 3: Möbius

References in general: make sure that there are spaces before parentheses.

Response to Comments of Reviewers

Response to Comments of Reviewer 1

Reviewer 1's general comments: In the present manuscript, the author introduced the synthesis of Doubly N-Confused and Ring-Contracted [24]Hexaphyrin Pd-Complexes and investigated the aromaticity using of the NICS method.

In general, aromatic compounds have high thermodynamic stability and are substantially more stable than antiaromatic compounds. Various indices of aromaticity exist, however as we know, the concept of aromaticity is complicated and a single value alone is often not enough to define a global property of the molecule.

There are many other reports that the NICS values calculated at the ring centers of polycyclic π systems do not always reflect the aromaticities of the rings. See, for example:

- 1) J. Aihara, Incorrect NICS-Based Prediction on the Aromaticity of the Pentalene Dication, *Bulletin of the Chemical Society of Japan*, 2004, 77, 101-102.
- 2) S. Fias, FV Damme, P. Bultinck, Multidimensionality of Delocalization Indices and Nucleus Independent Chemical Shifts in Polycyclic Aromatic Hydrocarbons, *Journal of Computational Chemistry*, 2008, 29, 358-366.
- 3) S. Fias, FV Damme, P. Bultinck, Multidimensionality of Delocalization Indices and Nucleus-Independent Chemical Shifts in Polycyclic Aromatic Hydrocarbons II: Proof of Further No locality, *Journal of Computational Chemistry*, 2010, 31, 2286-2293.
- 4) P. Bultinck, S.Fias, R. Ponec, Local Aromaticity in Polycyclic Aromatic Hydrocarbons: Electron Delocalization versus Magnetic Indices, *Chemistry-A European Journal*, 2006, 12, 8813-8818.
- 5) J. Aihara, H. Kanno, Local Aromaticities in Large Polyacene Molecules, *Journal of Physical Chemistry A*, 2005, 109, 3717-3721.
- 6) J. Aihara, Nucleus-independent chemical shifts and local aromaticities in large polycyclic aromatic hydrocarbons, *Chemical Physics Letters*, 2002, 365, 34-39.
- 7) J. Aihara, Spherical aromaticity in fullerenes and the nucleus-independent chemical shifts at the cage centers, *Bulletin of the Chemical Society of Japan*, 2003, 76, 103-105.

8) J. Poater, I. Garca-Cruz, F. Illas, M. Sola, Discrepancy between common local aromaticity measures in a series of carbazole derivatives, *Physical Chemistry . Chemical Physics*, 2004, 6, 314-318.

9) A. Kerim, Aromaticity and kinetic stability of fullerene C₃₆ isomers and their molecular ions, *Journal of Molecular Modeling*, 2011, 17, 3257-3263.

10) J. Aihara, Re-interpretation of the 2(N+1)2 rule for polyhedral closed-shell π -systems, *Chemical Physics Letters*, 2003, 375, 571-575.

Thus, NICS are very problematic for different reasons and such a comparison is not trustable.

The manuscript contains some scientific errors:

Page 6, line 123: “**8** and **9** are the first example of stable antiaromatic N-confused expanded porphyrins”.

I do not entirely agree with the statements.

In view of my critical comments listed above, my recommendation is to reject this manuscript for publication as an article in *Nature Communications*.

Response: We agree with **reviewer 1** that the (anti)aromaticity is complicated and cannot be simply judged by NICS values. Thus many other types of data are collected though the calculation of NICS values is one of the most popular methods to judge the (anti)aromaticity of porphyrinoids. The NMR shifts of beta-Hs in high field region also indicate that these expanded porphyrins are weakly antiaromatic. In addition the ACID diagram and HOMA value of these compounds are calculated and presented, both results are consistent with their NICS values. Therefore, these hexaphyrins are proved to be the first examples of stable antiaromatic N-confused expanded porphyrins.

Response to Comments of Reviewer 2

Reviewer 2's general comments: The article describes the synthesis of several novel porphyrinoids from the palladium-mediated coupling of 1,14-dibromotripyrrin **3b**. Homocoupling of **3b** afforded doubly N-confused contracted hexaphyrin Pd(II) complexes, while hetero-coupling of **3b** with dibenzylideneacetone yielded aromatic triphyrin **6**. The products were fully characterized, and the X-ray diffraction analysis nicely elucidated their structures. The aromaticity and antiaromaticity of compounds **6** and **8** were investigated based on ¹H NMR analysis and DFT

calculations. The authors obtained brominated hexaphyrins through regioselective bromination of **8**, which were employed to synthesize dimer **10** and porphyrin-appended hexaphyrin **12**.

(1) Reviewer 2 wrote: Doubly N-confused contracted hexaphyrin Pd(II) complex **8** is more appealing than other molecules because **8** is the first example of stable antiaromatic N-confused expanded porphyrins. To focus on this intriguing novel hexaphyrin, the authors should eliminate Scheme 2.

Response: To highlight the original purpose of this work, we prefer to keep Scheme 2 (now changed to Fig 2). We believe this may be friendlier for readers.

(2) Reviewer 2 wrote: In addition, the choice of reagents in Scheme 3 seems unreasonable. To avoid the formation of undesirable by-products from dba, the authors should use palladium complexes without the dba ligand, such as Pd(PPh₃)₄ and allylPdCp, to improve the yield of compound **8**.

Response: We already examined various Pd catalysts without dba ligand. Use of palladium complexes such as Pd(OAc)₂, Pd(PPh₃)₄ and Pd(PPh)₂Cl₂ led to much lower yields of **8**. According to the reviewer's suggestion, allylPdCp was also applied to the reaction but no target molecule was found.

(3) Reviewer 2 wrote: Moreover, the formation of **8** from dibromotripyrrin **3b** is fascinating from the mechanistic viewpoint. Unfortunately, the possible reaction mechanism is not discussed in the present manuscript.

Response: Due to the limit of the length of the main manuscript, the possible reaction mechanism was discussed in supplementary information in the first submission.

(4) Reviewer 2 wrote: Regioselectivity in the bromination of compound **8** should also be discussed.

Response: We calculated the HOMO orbital of **8** but could not find large coefficients at the brominated position. Thus, it is considered that the regioselectivity of bromination is mainly determined by steric hindrance.

(5) The purpose of the synthesis of compounds **10** and **12** is not clear. The formation of these molecules is not very surprising, and the authors should demonstrate their unique properties originated from the intrinsic property of the contracted hexaphyrin core.

Response: We synthesized compounds **10** and **12** to demonstrated that DNCRC unit **8** can be used for cross coupling reactions. As discussed in the text, the waves due to the first oxidation potential are split in the dimer **10**. Many reversible redox waves are observed for the triad **12**. Remarkably the UV-Vis absorption spectrum of **10** indicates a slightly enhanced long tail over 1250 nm, probably due to the substantial electronic interaction among the DNCRC units.

(6) Reviewer 2 wrote: The manuscript would be much improved if the authors could find a better synthesis of the antiaromatic hexaphyrin palladium complex **8** and demonstrate its specific properties and functions.

Response: We tried to apply some other methods to synthesize complex **8** such as condensation strategy. Unfortunately, we have not achieved complex **8** by using these methods.

Response to Comments of Reviewer 3

Reviewer 3's general comments: This is an interesting and insightful study on the synthesis of N-confused expanded porphyrins and an aromatic keto-triphyrin. Pd catalyzed cross-coupling of a diboryltripyrrane (or tripyrrin?) with 2,4-dibromoanisole gave a contracted benziporphyrin together with a tripyrrolic byproduct. At first glance it is hard to see where this comes fro but the structure results from incorporation of dibenzalacetone from the catalyst Pd₂(dba)₂. The keto-triphyrin is aromatic due to polarization of the keto-moiety. The calculated NICS value is -12.15 ppm. This is an interesting system but the authors should note that related keto-carbaporphyrins have been reported (Inorg. Chem. 2017, 56, 11426; Org. Chem. Front. 2022, 9, 5440). In carbaporphyrin systems the aromatic circuit competes with antiaromatic pathways resulting in weakly aromatic structures. However, in a related chlorin, the antiaromatic pathway is no longer a possibility and as is the case here the structure becomes strongly aromatic. Initially, the authors obtained a low yield of the keto-triphyrin but a more targeted approach raised the yield to 8.8%.

Response: We thank the suggestion of **Reviewer 3**. We added references of related keto-carbaporphyrins.

(1) Reviewer 3 wrote: As a slight point of confusion, the authors refer to 3a as a trippyrrin but it is shown in Scheme 2 as a tripyrrane. In Scheme 3, structure 3b is written as a tripyrrin, Is structure 3a a mistake?

Response: the structure of **3a** is correct, but the name of **3a** is wrong. We revised “ α,α' -diboryltripyrrin” as “ α,α' -diboryltripyrrane”.

(2) **Reviewer 3 wrote:** Pd catalyzed coupling of dibromotripyrrin **3b** gave two hexaphyrin products together with the keto-triphyrin and an unusual tripyrrolic palladium complex. The byproducts both originated from incorporation of dibenzalacetone from the catalyst. Hexaphyrin **8** was isolated as a bis-Pd(II) complex and has two N-confused pyrrole units. The system essentially have two carbacorrole-like binding pockets. Product **9** is a similar "contracted" hexaphyrin but is a mono-palladium complex. In this case, one of the confused rings has been oxidized to give an internal keto-unit. Compound **9** can be further metalated to give bis-Pd, Pd-Ni and Pd-Cu complexes. This is an interesting series of metalated doubly-confused porphyrinoids and importantly the possibilities for metalation are considerably altered compared to the doubly N-confused hexaphyrins previously reported by Furuta. Bromination of the bis-Pd(II) complex, followed by coupling with Ni(cod)₂ gave a selectively linked dimer. In addition, dibromination, followed by Suzuki-Miyaura cross-coupling with a nickel(II) porphyrin gave a diporphyrinylhexaphyrin triad. Excellent yields are reported. The electronic absorption spectra indicate that there are significant electronic interactions. In addition to spectroscopic and electrochemical characterization, X-ray crystal structures of **6**, **7**, **8**, **9**, **11** and **12** are provided. The only thing that seems to be missing is carbon-13 NMR. Is there a reason that carbon-13 NMR spectra are not included? While these will give limited information compared to proton NMR spectroscopy, I would have expected these to be included. If the samples are not sufficiently soluble to obtain this data, this should be noted.

Overall, this is an excellent study. I found a few minor errors but otherwise that is about it.

Response: We added the ¹³C NMR spectra of compounds **6**, **7**, **8**, **9**, **9Pd** and **9Ni** to SI. Unfortunately, the solubilities of compounds **10** and **12** are too poor to obtain the ¹³C NMR spectra with reasonable I/sigma.

(3) **Reviewer 3 wrote:** Page 1: "new comers" should be "newcomers", although after 30 years I am not sure NCPs can still be considered to be newcomers any more.

Response: Corrected as suggested, thanks.

(4) **Reviewer 3 wrote:** Line 104: "NMR spectrum" rather than just "NMR".

(5) **Response:** Corrected as suggested, thanks.

(6) Reviewer 3 wrote: Line 108: a NICS value of 5.41 is somewhat borderline for antiaromaticity.

Response: We agree with the viewpoint of Reviewer 3. We rephrased this sentence as “the NICS value at the center of molecule was calculated to be 5.41, indicating the weak antiaromatic character of **8**.”

(7) Reviewer 3 wrote: Line 131 and elsewhere: 5.40~4.09 ppm and 5.20~3.77 ppm should be 5.40-4.09 ppm and 5.20-3.77 ppm. These are ranges not approximations.

(8) Response: Corrected as suggested, thanks.

(9) Reviewer 3 wrote: Experimental section: column chromatography was performed using CH₂Cl₂/*n*-hexane but the authors never provide the ratio of the solvents. Was it 1:1 or something else entirely? Whatever the proportions, and these are likely to vary from experiment to experiment, the ratios should be provided.

Response: We added the ratio of the eluents to manuscript.

(10) Reviewer 3 wrote: Reference 3: Möbius

References in general: make sure that there are spaces before parentheses.

Response: Corrected as suggested, thanks.

Reviewers' Comments:

Reviewer #1:

Remarks to the Author:

[Note from the Editor: Please see attached files]

In the present manuscript, the author reported the synthesis of Doubly N-Confused and Ring-Contracted [24]Hexaphyrin Pd-Complexes and investigated the aromaticity. The conclusion is indeed interesting. I recommend this manuscript for publication in Nature Communication.

Ablikim Kerim

Reviewer #2:

Remarks to the Author:

In the revised manuscript, the authors included the ring current visualization of compound 8 by the ACID analysis to evaluate its antiaromatic character. Because the NICS analysis is sometimes problematic in investigating the aromaticity/antiaromaticity of pi-conjugated molecules, it was nice to add the results of the ACID analysis.

However, I was quite disappointed because I could not find much advancement in the revised manuscript in other aspects. The synthesis of the present molecules has not been improved. Moreover, the selective bromination of 8 and the preparation of porphyrin arrays would be interesting for specialists working in porphyrin chemistry but do not warrant publication in high-impact general journals such as Nature Communications.

Reviewer #3:

Remarks to the Author:

This is a revised manuscript and I will not repeat my earlier comments. This is an interesting submission that reports the synthesis of novel porphyrinoid structures. A diborylated tripyrrane reacted with 2,4-dibromoanisole to give a contracted benziporphyrin and a keto-triphyrin. The latter structure, which arises from reaction with the catalyst, has global aromatic character arising from the presence of a polarized carbonyl moiety. Reaction of a related dibromotripyrrane in the presence of the same Pd catalyst afforded contracted doubly N-confused hexaphyrins. The macrocycles have antiaromatic character. Although one of the reviewers was not satisfied with the description of these structures as being antiaromatic, the authors do provide solid support for this designation. NICS by itself would certainly not be enough but the proton NMR data clearly indicate the presence of a paratropic ring current. Other computational results are also provided. The authors have made a good effort to correct the manuscript and to take the referees' comments into account. There are a couple of minor points to be made. Although the authors correctly describe the intermediates as tripyrranes in the main text, they still describe them incorrectly as tripyrrins in the experimental section. On page 6, the sentence on line 8 might be reworded. Suggestion: "impeded a detailed analysis of the structure".

Response to Comments of Reviewers

Response to Comments of Reviewer 1

Reviewer 1's general comments: In the present manuscript, the author reported the synthesis of Doubly N-Confused and Ring-Contracted [24]Hexaphyrin Pd-Complexes and investigated the aromaticity. The conclusion is indeed interesting. I recommend this manuscript for publication in Nature Communication.

Response: We thank the evaluation of **Reviewer 1**.

Response to Comments of Reviewer 2

Reviewer 2's general comments: In the revised manuscript, the authors included the ring current visualization of compound 8 by the ACID analysis to evaluate its antiaromatic character. Because the NICS analysis is sometimes problematic in investigating the aromaticity/antiaromaticity of pi-conjugated molecules, it was nice to add the results of the ACID analysis.

Response: We thank the evaluation of **Reviewer 2**.

Reviewer 2 wrote: However, I was quite disappointed because I could not find much advancement in the revised manuscript in other aspects. The synthesis of the present molecules has not been improved. Moreover, the selective bromination of 8 and the preparation of porphyrin arrays would be interesting for specialists working in porphyrin chemistry but do not warrant publication in high-impact general journals such as Nature Communications.

Response: We accept the comments of the reviewer 2 partly and we cannot agree with him completely. We prepared β -pyrolyl dipyrin, however, the condensation of this intermediate with aldehydes in the presence of acid did not lead to the target product, only linear side products were obtained. We think at present only the way described in this work could synthesize the target molecule successfully. Although the yield is not so high, it could synthesize the target molecule with highly repeatable. Which is very important for a new way to make novel porphyrinoids.

Response to Comments of Reviewer 3

Reviewer 3's general comments: This is a revised manuscript and I will not repeat my earlier comments. This is an interesting submission that reports the synthesis of novel porphyrinoid structures. A diborylated tripyrrane reacted with 2,4-dibromoanisole to give a contracted

benzporphyrin and a keto-triphyrin. The latter structure, which arises from reaction with the catalyst, has global aromatic character arising from the presence of a polarized carbonyl moiety. Reaction of a related dibromotripyrrane in the presence of the same Pd catalyst afforded contracted doubly N-confused hexaphyrins. The macrocycles have anti aromatic character. Although one of the reviewers was not satisfied with the description of these structures as being antiaromatic, the authors do provide solid support for this designation. NICS by itself would certainly not be enough but the proton NMR data clearly indicate the presence of a paratropic ring current. Other computational results are also provided. The authors have made a good effort to correct the manuscript and to take the referees' comments into account.

Response: We thank the evaluation of **Reviewer 3**.

(1) Reviewer 3 wrote: There are a couple of minor points to be made. Although the authors correctly describe the intermediates as tripyrranes in the main text, they still describe them incorrectly as tripyrrins in the experimental section.

Response: Corrected as suggested, thanks.

(2) Reviewer 3 wrote: On page 6, the sentence on line 8 might be reworded. Suggestion: "impeded a detailed analysis of the structure".

Response: Corrected as suggested, thanks.